# Educational interventions to improve pediatric emergency care: A qualitative assessment of the perspectives of African healthcare workers

Emily A. Hartford[1]*, Chris A. Rees[2], Isaac Kihurani[3], Syeda Ra'ana Hussain[3], Elena Seifert[4], Alexis Schmid[5], Tigist Bacha[6], Carol C. Chen[7], Megan L. Schultz[8]

**1** Division of Emergency Medicine, Department of Pediatrics, University of Washington, Seattle, Washington, United States of America, **2** Division of Pediatric Emergency Medicine, Emory University School of Medicine and Children's Healthcare of Atlanta, Atlanta, Georgia, United States of America, **3** Paediatrics, Aga Khan University Hospital, Nairobi, Kenya, **4** Medical College of Wisconsin, Milwaukee, Wisconsin, United States of America, **5** Global Health Program and Emergency Medicine, Boston Children's Hospital, Boston, Massachusetts, United States of America, **6** Paediatric Emergency Medicine and Critical Care, Saint Paul's Hospital Millennium Medical College, Addis Ababa, Ethiopia, **7** Division of Pediatric Emergency Medicine, Department of Emergency Medicine, University of California San Francisco, San Francisco, California, United States of America, **8** Section of Pediatric Emergency Medicine, Department of Pediatrics, Medical College of Wisconsin, Milwaukee, Wisconsin, United States of America

\* Emily.hartford@seattlechildrens.org

## Abstract

Pediatric emergency care (PEC) training for health care workers (HCWs) is commonly offered in the form of short courses. This study gathers the perspectives of HCWs from eight African countries on how to best deliver and implement short training courses in PEC. This is a qualitative study using semi-structured key informant (KI) interviews. Utilizing the African Federation for Emergency Medicine (AFEM) member list serve, we identified African HCWs who had previous experience participating in and/or delivering short training courses in PEC. From this cohort, four interviewers were selected. These interviewers all received training in qualitative interviewing and then each recruited five KIs in their respective settings using convenience sampling. All interviews were recorded, transcribed, translated as necessary, and coded using thematic analysis. A total of 20 interviews were completed. Most KIs (75%) were physicians. Several themes on short training courses in PEC emerged: there was strong motivation to participate in PEC trainings, interactive sessions were preferred over didactic sessions, the recommended course structure was a half-day format with longitudinal follow-up, and the ideal for course instructors was a mix of local trainers and visiting trainers. KIs reported several potential negative consequences of short training courses in PEC, including clinical staffing gaps during courses and PEC content taught that was incongruous with local protocols. Future curricular development and implementation of short training courses in PEC should incorporate the preferences and best practices identified by African HCWs, namely interactive sessions with longitudinal follow-up given by a mix of local and visiting trainers. Our study limitations include the number of participants and potential for selection bias.

**Data availability statement:** Data is available on Open Science Framework at https://osf.io/aeq8b/?view_only=c926e978de1c-42f78a11347932eca15d

**Funding:** This study was supported by funding from the Global Health Research Seed Project Funding for Faculty, Medical College of Wisconsin Office of Global Health (to author MS). One author received a small stipend from this grant for completing interviews (TB). One author receives salary support from the US National Institutes of Health (K23HL173694 to CR). The funders had no role in the study design, data collection and analysis, decision to publish, or preparation of the manuscript. There was no other salary support for authors for this study.

**Competing interests:** The authors have declared that no competing interests exist.

## Introduction

Despite ongoing progress, sub-Saharan Africa still has the highest mortality rate in the world for children under five years at 74 per 1,000 children [1]. This is approximately 16 times higher than the average for high-income countries (HIC) [1,2]. In low- and middle-income countries (LMICs), a high proportion of these deaths occur within the first 24 hours of presentation to a healthcare facility [3–5]. Initiatives to improve early diagnosis and timely resuscitation among critically ill children have been shown to reduce mortality rates in some settings in sub-Saharan Africa [6,7]. Therefore, specialized training in PEC for HCWs may play a key role in decreasing child mortality in LMICs.

Residency and fellowship programs for PEC specialists are developing in Africa, but still scarce [8]. In the interim, short courses on various pediatric and emergency skills have been used to increase capacity for PEC, taking into consideration the clinical demands of many African HCWs. These short training courses in PEC cover a wide range of topics, from neonatal resuscitation to procedural sedation [9–12]. When studied, these courses often assess participant perceptions or short-term knowledge gains using pre- and post-assessments [13–15]. However, there is a lack of knowledge about preferred course format, modality, and timing, or the potential negative consequences of such short courses in LMICs.

The AFEM was founded in 2009 with the goal of achieving high-quality emergency care for patients across Africa [16]. In 2016, a working group of clinician volunteers was recruited to develop an open-source curriculum for PEC training courses. The resulting curriculum has been piloted using various formats in multiple countries, with participant feedback demonstrating a need for more information regarding the ideal format, modality, and instructors for PEC training. The objective of this study is to gain insight from African HCWs on their preferred methods for learning PEC through short courses.

## Methods

### Study design

This is a qualitative study using semi-structured interviews of African HCWs to identify the preferred educational formatting for short training courses in PEC. Short training courses in PEC were defined as topic-driven, focused on a pediatric or emergency medicine subject, with a duration of days to weeks. This study was approved by the institutional review boards of Seattle Children's Hospital (STUDY00002699), University of California San Francisco (Ref 295959), Boston Children's Hospital (IRB-P00037537), Madison College of Wisconsin (PRO00039214), and Aga Khan University (Ref 2021/IERC-44(v1)). Verbal informed consent was obtained from all key informants prior to completing the interviews.

### Interviewer and participant recruitment

A recruitment email was sent to all AFEM members in February 2020 to identify collaborators interested in a qualitative assessment of pediatric emergency training (S1 Appendix). The AFEM has >2,000 members who are based in 55 countries [16]. Of the respondents, the study team selected four interviewers based on regional representation and experience with qualitative research. All four interviewers participated in a 60-minute training session with the authors about interviewing and reflexivity in qualitative research [17]. Interviewers were 2 consultant physicians in pediatric emergency and critical care, one consultant in PEC and 1 nurse with experience working in PEC. They were each asked to recruit five KIs from their colleagues with experience providing PEC in sub-Saharan Africa and either teaching or participating in short training courses in PEC. Interviewers were encouraged to identify KIs from

diverse backgrounds, including profession, gender, age, and site of practice. They approached their own co-workers and individuals within their professional networks to recruit a diverse group of KIs with experience in participating in or teaching PEC short courses. Interviewers did not report difficulty with identifying participants to complete the interviews. Each interview began with a scripted statement about the voluntary nature of their participation and by obtaining verbal consent for participation and recording/transcribing the interview (S2 Appendix). The interview duration was approximately one hour.

### Interview content development

Interview questions were developed by a group of specialists in PEC and medical education after the early implementation and piloting of AFEM training materials in LMICs [18]. Question prompts were organized into the following sections: preferred educational format, preferred schedule, preferred characteristics of instructors, incentives and deterrents to participation, and benefits and potential negative impacts of short training courses. The interview tool was piloted and refined amongst the authors and with additional nursing and physician colleagues based in sub-Saharan Africa (S2 Appendix).

### Study procedures

Interviews were conducted virtually and recorded in the participant's preferred language using Zoom or Skype, according to participant preference. Interviewers and KIs were financially compensated (USD $25 for KIs, $250 for interviewers). French interviews were translated to English by certified professional translators and all interviews were manually transcribed using the online platform www.transcribe.com by author ES. All recordings and transcriptions were stored in an encrypted data storage repository in Dropbox. Interviews were de-identified prior to being uploaded for analyses.

### Analyses

Using Braun and Clarke's six-phase framework for thematic analysis [19], authors AS and MS individually familiarized themselves with the datasets, created open codes, and searched for themes and then defined the themes together to make an initial codebook. The codebook was tested and iteratively refined until inter-rater reliability scores among the coders achieved Cohen's kappa coefficient 0.80. Authors EH and ES coded all 20 interviews, resulting in each interview being coded twice. Authors IK and CR independently performed thematic analysis and then met to discuss overlapping themes and achieve consensus after discussion of relevance and veracity of identified themes. A conventional content analysis approach was used to identify recurrent and salient themes. All analyses were conducted using the qualitative analysis software Dedoose (version 9.0.46 Los Angeles, CA: SocioCultural Research Consultants, LLC).

## Results

A total of 18 AFEM members responded with interest in participating in this study; four interviewers were selected. The remainder of interested respondents were asked if they would like to participate in the study as KIs. Interviewers completed a total of 20 interviews between May 5 and June 30, 2021.

Most KIs were medical doctors (15/20; 75%), three were nurses, two were clinical officers (Table 1). Most participants had pursued additional training beyond medical school or their main certification course (13/20; 65%); the most common advanced training program was a pediatric residency (9/13; 69%). Three participants completed other post-graduate, non-residency training, and two had additional doctoral degrees.

**Table 1. Key informant characteristics.**

| Type of health care worker | n (%) |
|---|---|
| Medical Doctor | 15 (75%) |
| Nurse | 3 (15%) |
| Clinician, non-MD | 2 (10%) |
| Years of Experience | 10 median (range 1–34, IQR 4.5–17) |
| Country of Practice | |
| Burkina Faso | 1 |
| Burundi | 4 |
| Ethiopia | 5 |
| Kenya | 3 |
| South Africa | 4 |
| South Sudan | 1 |
| Tanzania | 1 |
| Uganda | 1 |

There was a wide range of clinical experience among KIs, from less than one year in practice to 34 years. The median experience level was 10 years (IQR 4.5–17 years). The majority of KIs who answered a question about their practice site reported working in a public hospital (14/17;82%), four reported working in a private hospital, one worked in both settings. Most KIs who reported hospital type worked in higher-level settings: 16/19 (84%) worked in tertiary care centers, one worked in a secondary care setting, and two worked in district hospitals. The geographic representation of KIs was mostly Eastern African, with 14/20 (70%) practicing in Uganda, Burundi, Kenya, Ethiopia, or Tanzania. A total of eight countries were represented.

## Preferred characteristics of instructors for short courses

Key informants wanted instructors who had sufficient experience to teach on the specific PEC topics (Table 2). They favored a mix of local instructors with visiting instructors, emphasizing the importance of local instructors for clinical relevance and follow-up guidance. There was also some discussion about the importance of instructors being open to feedback from students.

## Preferred educational format for short courses

When asked about preferred learning modality for short training courses in PEC, most common response was any type of interactive training, including patient cases, group discussions, and simulations (Table 2). Participants specifically expressed a preference for interactive training over traditional didactic lectures. There was also interest in the provision of curricular resources for local HCWs to use, and for online learning through lectures or self-guided courses. Key informants were in favor of train-the-trainer (ToT) models of instruction for wider dissemination, as well as ongoing training sessions to support knowledge retention.

## Schedule for short courses

Regarding preferred timing of short courses, there was a preference among KIs for longitudinal training over time (Table 2). They preferred the schedule of an initial intensive session with the opportunity for follow up to enhance knowledge retention. Key informants also supported limiting short course duration to one half-day of training at a time so participants could rest or complete their clinical work.

**Table 2. Thematic results from qualitative interviews to determine preferred format, modality, and instructors for curriculum for training in pediatric emergencies.**

| Theme | Sub-themes | Frequency | Representative quote |
|---|---|---|---|
| **Preferred training modalities** | | | |
| **Characteristics of Instructors** | Trainer should have sufficient experience to train | 65 | "For some topics, a train the trainer model would be fine. But then you need the specialist to come and teach them more serious topics or a bit more detail." |
| | Preference for local trainers | 24 | "I would say a local trainer would be better, especially when they understand our 'on the ground' in terms of the equipment and the setting that we have. I think it would be more realistic for the training." |
| | Preference for foreign trainers | 12 | "I think for the, for the biggest part I would prefer local people because they know your, understand your challenges very well, but definitely to be, to be balanced with some visiting teachers. There is a saying that 'a prophet is not known in his own country', something like that. But I think people just listen to, to, people just give more attention to someone that comes with some credential, that comes from a different country. It is as if people put more value to the information that they give if I can say it like that. And it also, it adds some credit to your course that you are doing. So I think there needs to be a balance between the two." |
| | Trainers open to feedback | 10 | "I think the more you can provide and get feedback closer to the people that you're teaching, uh so a local supervisor… I think that's essential and so the smaller you can make that, or the bigger you can make the pool of trainers and the smaller you can make that group of trainees, the better." |
| **Preferred Educational Format** | Interactive Training, Case Discussions | 250 | "Case discussions and group work provide a good learning environment in the sense that what is acquired from the sessions or what was prepared by the lecturers there is room for more interactions. In very many instances you find that matter helps in the long run. That was one of my great learning modalities." |
| | Interactive Training, Simulations | 119 | "Really what I like is the simulation-based training because simulation just simulates. We assume the case is happening so we can learn so many things from simulation. As emergency physician, in order to be experts you have to practice on a mannequin before you practice on a patient, so simulation-based teaching is very important." |
| | Lectures Less Preferred Compared to Interactive Training | 87 | "The training model should be in such a way that it should be more practical than theoretical. That is what I would encourage. More case discussion than lectures with slides. It should be more about demonstrations and interactive learning." |
| | Curricular Resources Provided | 81 | "They're usually preformed lectures and course content that inform manuals that are provided to assist during training sessions. and they remain constant unless there is an updated version. These would be helpful for an instructor to teach others and to students." |
| | Online learning (e.g., lectures, self-guided courses) | 81 | "Some available resources that are pre-recorded and available online that people go through and they come to the course with a lot of theory is ideal. Also, a pre-course assessment that gives you a sense of your knowledge. In fact, that assessment gives them feedback as to where their knowledge gaps are and also gives the instructor an idea of where they are." |
| | Promoting Knowledge Retention | 50 | "I think there should be a continuous reading before and after the training so it would help to strengthen the knowledge and skills. Also, to be involved actively in management of pediatric emergency patients would help enormously to retain knowledge." |
| | Train the trainer cascade dissemination strategies | 65 | "A train the trainer model would be ideal, so that we could have one person who is an expert or a champion in that particular field, and then he should be training the rest of the colleagues in the facility." |
| **Training schedules** | Longitudinal training preferred over short-term training | 145 | "As compared with the short-term, the long-term one will be preferred because it would give you an opportunity to gain a good practical experience on a particular case and through challenges and repeated exposure." |
| | Duration of training sessions | 43 | "From my experience, what worked for us was giving a didactic class practical training for a week, and of course maybe the training for emergency and critical care may be a little bit longer, I think it should be."<br>"The lecture classes in a half-day. Because the fact that you are given time to rest, it is really advantageous for the participants. Therefore, the fact that you are returning in two or three …well, after taking a rest, after this type of day you need to rest for work." |

## Perceived incentives and deterrents to participating in short training courses

When KIs were asked about perceived incentives and deterrents to participating in short training courses, they mentioned that a financial incentive is helpful, but also that career development is important (Table 3). Deterrents included a lack of necessary equipment, either for the training itself, or at their clinical facilities. They also mentioned the cost of training,

**Table 3. Thematic results from qualitative interviews to understand incentives and deterrents to short course training.**

| Theme | Sub-themes | Frequency | Representative quote |
|---|---|---|---|
| **Incentives and deterrents to participate in trainings** | | | |
| **Incentives to Participate in Trainings** | Financial Incentive | 21 | "If staffs are provided with some financial incentives (for the training) then they may stay as well. So the two, the two are important. Their commitment, willingness, their heart for that particular training and also to support them (for the training) also." |
| | Career development | 8 | "The clinician perceives the training itself as an incentive for their career, the way they look at it... inside our country, they keep asking for regular training so this is the incentive that we have." |
| **Deterrents to Training** | Equipment requirements in training | 119 | "Obviously we are in a low-resource setup so we might not have the adequate instrumentation, we might not have the adequate drugs available, and so on to use what we learn in the training or to do the training." |
| | Cost of training | 36 | "So building this initial human resource is hard because of the fact that they have to pay their tuition... and then the day after they have the paper which they can only use in private practice." |
| | Internet related challenges | 24 | "This kind of learning, it needs connectivity, strong connectivity. So bad connectivity, technology, knowing also how to Zoom in or how to do that is also a challenge for trainings." |
| | Training venues, infrastructure | 19 | "So, in a situation like in a very poor country, might be very difficult to do simulation-based training. But the challenge is technology, mannequins, and maybe simulation room, which may be lacking." |
| | Training that is applicable to local setting | 48 | "I would say if it's something, if it's something that is simple and applicable and practical to us, then it's something that you can relate to within your own setting and then you know, yes that is something I will see in the future so that's what I'm going to focus on that and remember and that-or it's something that I've already seen so I think if it's like a practical and applicable, the type of topics and the way it's being presented, then I would remember it more." |
| | Administration values educational experience gained | 29 | "I am also looking at some gaps toward the policy makers and the, the government officials such as the ministry of health. Like if you take pediatric emergency is now-we are too full. And administration of maternal and child health often falls to adult critical care directors. So I can see that there is a huge gap from the policy makers for this issue so training can help." |

lack of internet, and lack of physical classroom space as additional barriers to participating in short training courses.

## Perceived benefit of short training courses

Regarding the benefits of participating in short training courses in PEC, many described intrinsic value for themselves as HCWs and a desire to improve their patient care with new pediatric skills and knowledge (Table 4). They emphasized the importance of training that is relevant for their setting. There was also mention of administrators valuing the educational experiences gained by short course participants.

## Potential negative impacts of short training courses

Many KIs were concerned about decreased quality of clinical care while trainings were occurring because crucial HCWs were absent from their clinical duties during short courses. A few KIs mentioned the ability to create a schedule that preserved clinical staffing during a short training course may mitigate loss of clinical workforce. There was also discussion of irrelevant material shared by visiting trainers, although most appreciated learning about PEC elsewhere and how it could be applied locally as well.

## Discussion

In this qualitative study of 20 HCWs in sub-Saharan Africa, we found KIs favorably perceived participation in short training courses in PEC. There was a strong preference for interactive training over didactic lectures, and KIs wanted to learn from both local and visiting trainers. They recommended limiting sessions to a half-day to preserve clinical staffing, as well as starting a short course with an intensive session followed by longitudinal refreshers. They also favored a ToT model and the provision of curricular materials for HCWs to utilize locally.

**Table 4. Thematic results from qualitative interviews to determine perceived impact of short course trainings.**

| Theme | Sub-themes | Frequency | Representative quote |
|---|---|---|---|
| **Perceived impact of trainings** | | | |
| **Perceived Benefits of Trainings** | Participant perceives value in the training | 147 | "And you know, there are certain things that we have never seen here and it is very difficult to teach those kinds of topics. Certain spider bites or snake bites that we don't have here or things like that, you know, there might be topics that we are blind to. And so I think there's definitely value to get someone from outside to give another perspective, you know, to see, you know, what other people are doing and to see if you are current approaches are accurate." |
| | Sustainable knowledge retention and skill improvements | 74 | "We need continuous training because we will always need to improve our knowledge and skills. It isn't just us … but also others, to repeat it to the younger doctors who are behind us. Because we realize that most children die in the pediatric emergency rooms and especially, that it is due to a lack of training. Therefore, we don't lack the situations to push us to do it. The children are dying in our hands." |
| | Training that is applicable to local setting | 48 | "I would say if it's something, if it's something that is simple and applicable and practical to us, then it's something that you can relate to within your own setting and then you know, yes that is something I will see in the future so that's what I'm going to focus on that and remember and that-or it's something that I've already seen so I think if it's like a practical and applicable, the type of topics and the way it's being presented, then I would remember it more." |
| | Administration values educational experience gained | 29 | "I am also looking at some gaps toward the policy makers and the, the government officials such as the ministry of health. Like if you take pediatric emergency is now-we are too full. And administration of maternal and child health often falls to adult critical care directors. So I can see that there is a huge gap from the policy makers for this issue so training can help." |
| **Potential Negative Impact of Trainings** | Missing clinical work due to training schedule | 70 | "When I am not there, this paramedical staff must take care of my patients and must take care of pediatric emergencies. You can imagine how the ward is really affected." |
| | Instruction supplants previously used safe practices | 40 | "When a professor or trainer who comes from the outside and is not used to our protocols, he comes and teaches things that really might conflict with protocols which already exist. |

Challenges to short training courses in PEC included lack of necessary equipment and shortage of staff during training courses.

There is a critical shortage of HCWs in sub-Saharan Africa, with an average of 0.2 physicians and 1 nurse per 1,000 people, in comparison to HICs with 3.1 physicians and 10 nurses per 1,000 people [20]. Sustainable models such as ToT, developed to promote wider dissemination of important clinical skills, particularly in resource-limited environments, remain a key aspect of capacity-building, strengthening clinical care, and improving patient outcomes. A systematic review of ToT models for health education showed significant improvement in participant knowledge and behaviors and evidence of regional benefit from training [21]. There is also a published framework for ToT programs that includes developing local master trainers, providing the appropriate resources for training, ensuring alignment of the course to local needs and resources, guiding effective implementation, and nurturing ongoing learning and development over time [22]. The main themes elucidated in this study aligned closely with these ToT concepts: KIs agree with the utilization of both local and visiting instructors to develop and implement interactive training sessions, with provision of necessary resources and follow-up training for sustainability.

This study revealed a strong preference for interactive training of any kind, including simulation, case-based learning, group discussion, and skills stations, over traditional didactic lectures. Interactive learning has been recognized as more effective in recent years in HICs as well, with major universities moving towards completely removing lectures from the medical school curriculum [23]. Two different reviews of active learning methodologies including case discussions, simulation, and practice with feedback, both reported favorable outcomes for patients and learners alike [24,25]. Another review of the 'flipped classroom' technique, where learners view videos before in-person groupwork and teaching, found increased knowledge retention when compared to traditional methods [26]. In an emergency training course in Malawi, authors found simulation to be helpful and feasible; instructors were able to shorten the course

and maintain the same level of knowledge using simulated scenarios [27]. Simulation-based training has also been shown to boost the confidence, knowledge, and skills of HCWs handling pediatric emergencies in Botswana [28] and Kenya [29]. Virtual simulation is another useful tool that has been successfully utilized for interactive distance learning [30]. Thus, across multiple settings with various resources, interactive teaching methods are preferred by participants and are associated with improved educational outcomes. Our results affirm this and add the participant desire for interactive learning in short training courses in PEC.

Key informants in our study were motivated to participate in additional PEC training to improve their own practice and care for pediatric patients. Although we did not specifically ask about preferred topics, KIs did mention a desire for enhancing their skills to care for acutely ill children. Pediatric resuscitation courses can be limited or expensive in resource-limited settings; a previous survey of African HCWs identified a gap in access to advanced life support training for approximately 25% of respondents across 23 countries [31]. Pediatric resuscitation is a core element of PEC training and would be beneficial to include in courses being offered to HCWs in LMICs. In Rwanda, there is a network of locally relevant training courses in resuscitation more broadly that could be used as a model for implementation elsewhere [32].

One novel goal of our study was to learn about the potential for negative consequences from short training courses. Training courses are a frequent part of academic partnerships and other donor-driven initiatives in LMICs, making it important to understand the trainee perspective on potential negative consequences. This study identified two related themes: trainings can interrupt clinical staffing, and trainings may contradict locally relevant protocols. The planning for any short training course in a resource-limited setting should take these concerns into account and work to mitigate potential negative effects. Partnering between visiting and local trainers on all aspects of the course content and implementation would likely help to avoid these negative consequences. These themes have not previously been described and are important to take into consideration when planning future courses.

There are several limitations of our study. We report the perspectives of 20 HCWs in eight African countries, but their views may not be fully representative given the huge variation in practice, culture, training, and resources across the continent. In addition, these eight countries primarily represented East Africa, so our findings may have limited generalizability. KIs may have responded with desirability bias in their answers, although interviewers specifically recruited their own colleagues to minimize this possibility. Selection bias is also possible, as those who responded to our initial AFEM email and those who agreed to be interviewed may have different views than the general HCW population. Finally, although we followed standards for qualitative analysis, there may have been bias in the identification of themes for reporting.

## Conclusion

In this qualitative study of 20 HCWs from eight African countries, we report the learning and timing preferences for short training courses in PEC. Participants were highly motivated to participate in short training courses in PEC, and they strongly preferred interactive teaching methods over didactic lectures. They desired experienced instructors with a mix of local and visiting trainers, and the ideal format was an intensive period of half-day sessions with more longitudinal follow-up. Key informants also encouraged the ToT model, provision of curricular materials for ongoing local use, and the need to mitigate gaps in clinical care during trainings. Future curricular development and implementation of short training courses in PEC should incorporate these preferences and best practices identified by African HCWs.

## Supporting information

**S1 Appendix. Recruitment Email.**
(DOCX)

**S2 Appendix. Interview Tool.**
(DOCX)

**S1 Checklist. Inclusivity in global research.**
(DOCX)

## Acknowledgements

We acknowledge the contributions of Dr. Prinetha Moodley, Ciza Bonne, and Dr. Adnaan Mustafa, who assisted with conducting qualitative interviews for this project. We would also like to acknowledge the leadership and coordination of AFEM in advancing educational content for improving pediatric emergency care.

## Author contributions

**Conceptualization:** Emily A. Hartford, Chris A. Rees, Isaac Kihurani, Syeda Ra'ana Hussain, Elena Seifert, Alexis Schmid, Carol C. Chen, Megan L. Schultz.

**Data curation:** Emily A. Hartford, Elena Seifert, Tigist Bacha, Megan L. Schultz.

**Formal analysis:** Emily A. Hartford, Chris A. Rees, Isaac Kihurani, Elena Seifert, Alexis Schmid, Tigist Bacha, Carol C. Chen, Megan L. Schultz.

**Funding acquisition:** Emily A. Hartford, Elena Seifert, Megan L. Schultz.

**Investigation:** Emily A. Hartford, Alexis Schmid, Megan L. Schultz.

**Methodology:** Emily A. Hartford, Isaac Kihurani, Syeda Ra'ana Hussain, Elena Seifert, Alexis Schmid, Megan L. Schultz.

**Project administration:** Emily A. Hartford, Elena Seifert, Megan L. Schultz.

**Supervision:** Emily A. Hartford.

**Writing – original draft:** Emily A. Hartford, Chris A. Rees, Isaac Kihurani, Elena Seifert, Alexis Schmid, Tigist Bacha, Megan L. Schultz.

**Writing – review & editing:** Emily A. Hartford, Chris A. Rees, Isaac Kihurani, Syeda Ra'ana Hussain, Elena Seifert, Alexis Schmid, Tigist Bacha, Carol C. Chen, Megan L. Schultz.

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
