## [Decision Letter · Decision Letter 0]

16 Jul 2024

PGPH-D-24-00876

Educational interventions to improve pediatric emergency care: a qualitative assessment of the perspectives of African healthcare workers

Dear Dr. Hartford,

Thank you for submitting your manuscript to PLOS Global Public Health. After careful consideration, we feel that it has merit but does not fully meet PLOS Global Public Health’s publication criteria as it currently stands. Therefore, we invite you to submit a revised version of the manuscript that addresses the points raised during the review process.

Your manuscript has been reviewed by two reviewers and their comments are available below. They have requested that additional details of the methodology are included as well as more variation in the included quotations. Please review these comments and revise the manuscript accordingly. 

We look forward to receiving your revised manuscript.

Kind regards,

Emma Campbell, Ph.D

Staff Editor

Journal Requirements:

1.  Please include a complete copy of PLOS’ questionnaire on inclusivity in global research in your revised manuscript. Our policy for research in this area aims to improve transparency in the reporting of research performed outside of researchers’ own country or community. The policy applies to researchers who have travelled to a different country to conduct research, research with Indigenous populations or their lands, and research on cultural artefacts. The questionnaire can also be requested at the journal’s discretion for any other submissions, even if these conditions are not met.  Please find more information on the policy and a link to download a blank copy of the questionnaire here:https://journals.plos.org/globalpublichealth/s/best-practices-in-research-reporting. Please upload a completed version of your questionnaire as Supporting Information when you resubmit your manuscript.

2. We do not publish any copyright or trademark symbols that usually accompany proprietary names, eg (R), (C), or TM  (e.g. next to drug or reagent names). Please remove all instances of trademark/copyright symbols throughout the text, including ® on page 17.

3. "In the online submission form, you indicated that Data can by accessed by request via a link to de-identified interview transcripts. 

a. In a public repository, 

b. Within the manuscript itself, or 

c. Uploaded as supplementary information.

This policy applies to all data except where public deposition would breach compliance with the protocol approved by your research ethics board. If your data cannot be made publicly available for ethical or legal reasons (e.g., public availability would compromise patient privacy), please explain your reasons by return email and your exemption request will be escalated to the editor for approval. Your exemption request will be handled independently and will not hold up the peer review process, but will need to be resolved should your manuscript be accepted for publication. One of the Editorial team will then be in touch if there are any issues."

Additional Editor Comments (if provided):

Reviewers' comments:

Reviewer's Responses to Questions

**Comments to the Author**

1. Does this manuscript meet PLOS Global Public Health’s publication criteria ? Is the manuscript technically sound, and do the data support the conclusions? The manuscript must describe methodologically and ethically rigorous research with conclusions that are appropriately drawn based on the data presented.

Reviewer #1: Yes

Reviewer #2: Yes

2. Has the statistical analysis been performed appropriately and rigorously?

Reviewer #1: Yes

Reviewer #2: Yes

3. Have the authors made all data underlying the findings in their manuscript fully available (please refer to the Data Availability Statement at the start of the manuscript PDF file)?

Reviewer #1: Yes

Reviewer #2: Yes

4. Is the manuscript presented in an intelligible fashion and written in standard English?

Reviewer #1: Yes

Reviewer #2: Yes

5. Review Comments to the Author

Reviewer #1: Thank you for the opportunity to review this article. I find it to be a very interesting and well-conducted qualitative research study. The findings are useful and reproducible in other low-resource countries. Although teaching through residencies is the ideal method, the insufficient number of these programs is a reality in many low- and middle-income countries. This adds interest to the research presented by the authors.

I believe the research successfully evaluates the perception of the short courses received and the reality in which they are implemented, which adds even greater value to the information obtained.

The reduction of staff during training sessions when the training frequency is not sustained and scheduled throughout the year, allowing protected time for training without affecting patient safety, is a very interesting point.

Although the authors consider the selection of respondents and their findings to be a limitation due to the sample obtained, I believe these are logical aspects that can be generalized to other low-income countries and continents. On the other hand, the selection of respondents likely gathered the opinions of providers involved and interested in learning and improving their skills; rather than being a limitation, these are probably individuals who have more tools to evaluate the items presented in the survey.

For all the above reasons, I consider that the research should be published.

My proficiency in English as a second language does not allow me to make stylistic corrections. I do not understand the following in line 139: author ES?

Reviewer #2: REVIEWER’S REPORT

TITLE: EDUCATIONAL INTERVENTIONS TO IMPROVE PEDIATRIC EMERGENCY CARE: A QUALITATIVE

ASSESSMENT OF THE PERSPECTIVES OF AFRICAN HEALTHCARE WORKERS

Thank you for giving me the opportunity to review this paper.

I commend the authors for contributing to the body of knowledge concerning educational interventions to improve paediatric emergency care in Africa. I reviewed the manuscript using the Consolidated Criteria for Reporting Qualitative Research (COREQ) guidelines: Tong A, Sainsbury P, Craig J. Consolidated criteria for reporting qualitative research (COREQ): a 32-item checklist for interviews and focus groups. International journal for quality in health care. 2007 Dec 1;19(6):349-57.

TITLE

Appropriate

ABSTRACT

The authors should include the study limitations in the abstract.

INTRODUCTION

The introduction provides enough background information for the study, and addresses why it should be done.

METHODOLOGY

Interviewer and Participant Recruitment

How were the key informants selected? Where there any challenges during the selection process or any reasons given for non-participation?

The relationship and extent of interaction between the four selected interviewers and their key informants should be described.

In addition, the personal characteristics of the interviewers and their assumptions or personal interests in the research topic, if any, should be stated.

Study Procedures

Please provide information about duration of the interviews.

Analyses

The use of multiple coders is commendable. Did the authors obtain feedback from key informants on the research findings?

RESULTS

Clearly presented. However, for supporting quotations, you may wish to include quotations from different participants.

DISCUSSION

Adequate.

General comments

This is an important study concerning educational interventions to improve pediatric emergency care within the region.

Please avoid beginning sentences with abbreviations. The authors may also wish to consider an English proofreading service.

Objective Evaluation: Accept after minor revisions are done.

6. PLOS authors have the option to publish the peer review history of their article (what does this mean? ). If published, this will include your full peer review and any attached files.

**Do you want your identity to be public for this peer review?** For information about this choice, including consent withdrawal, please see our Privacy Policy .

Reviewer #1: **Yes: ** viviana pavlicich

Reviewer #2: **Yes: ** Professor Ikenna Kingsley Ndu

---

## [Decision Letter · Decision Letter 1]

3 Dec 2024

Educational interventions to improve pediatric emergency care: a qualitative assessment of the perspectives of African healthcare workers

PGPH-D-24-00876R1

Dear Dr Hartford,

We are pleased to inform you that your manuscript 'Educational interventions to improve pediatric emergency care: a qualitative assessment of the perspectives of African healthcare workers' has been provisionally accepted for publication in PLOS Global Public Health.

Best regards,

Julia Robinson

Executive Editor

Reviewer Comments (if any, and for reference):

Reviewer's Responses to Questions

**Comments to the Author**

1. If the authors have adequately addressed your comments raised in a previous round of review and you feel that this manuscript is now acceptable for publication, you may indicate that here to bypass the “Comments to the Author” section, enter your conflict of interest statement in the “Confidential to Editor” section, and submit your "Accept" recommendation.

Reviewer #2: All comments have been addressed

2. Does this manuscript meet PLOS Global Public Health’s publication criteria ? Is the manuscript technically sound, and do the data support the conclusions? The manuscript must describe methodologically and ethically rigorous research with conclusions that are appropriately drawn based on the data presented.

Reviewer #2: Yes

3. Has the statistical analysis been performed appropriately and rigorously?

Reviewer #2: Yes

4. Have the authors made all data underlying the findings in their manuscript fully available (please refer to the Data Availability Statement at the start of the manuscript PDF file)?

Reviewer #2: Yes

5. Is the manuscript presented in an intelligible fashion and written in standard English?

Reviewer #2: Yes

6. Review Comments to the Author

Reviewer #2: The authors have addressed all my comments adequately, and the manuscript meets PLOS Global Public Health’s publication criteria.

7. PLOS authors have the option to publish the peer review history of their article (what does this mean? ). If published, this will include your full peer review and any attached files.

**Do you want your identity to be public for this peer review?** For information about this choice, including consent withdrawal, please see our Privacy Policy .

Reviewer #2: **Yes: ** Professor Ikenna Kingsley Ndu
